# Experimental Study on Dynamic Mechanical Properties of Sandstone Corroded by Strong Alkali

Qi Ping [1,2,3,*], Chen Wang [2,3], Qi Gao [2,3], Kaifan Shen [2,3], Yulin Wu [2,3], Shuo Wang [2,3] and Shijia Sun [2,3]

1    State Key Laboratory of Mining Response and Disaster Prevention and Control in Deep Coal Mine, Anhui University of Science and Technology, Huainan 232001, China
2    Engineering Research Center of Mine Underground Projects, Ministry of Education, Anhui University of Science and Technology, Huainan 232001, China; 2020200327@aust.edu.cn (C.W.); 2020200277@aust.edu.cn (Q.G.); 2020200341@aust.edu.cn (K.S.); 2020200404@aust.edu.cn (Y.W.); 2020200372@aust.edu.cn (S.W.); 2021200516@aust.edu.cn (S.S.)
3    School of Civil Engineering and Architecture, Anhui University of Science and Technology, Huainan 232001, China
*    Correspondence: ahpingqi@163.com or qping@aust.edu.cn; Tel.: +86-139-5645-9398

**Featured Application: The work is potentially applied to deep rock works in a long-term groundwater environment.**

**Abstract:** In order to analyze the effect of different corrosion times on the dynamic compression mechanical properties of sandstone, the coal mine sandstone specimens are subjected to corrosion in NaOH solution with pH 11 for 0 d, 1 d, 3 d, 7 d, 14 d, and 28 d, and then, the impact compression tests and Brazilian splitting test are conducted using a split Hopkinson pressure bar apparatus. The study results show that sandstone specimen mass and the average density growth rate increases, with the corrosion time first rapidly increasing and then tending to level off the trend. The impact of the compression specimens on the dynamic stress–strain curve change law is basically the same, but the time gradient curve shape is different, and the line elastic deformation stage and plastic deformation stage curve difference gradually decreases. The specimen dynamic compressive strength and the dynamic elastic modulus with corrosion time are quadratic, and the exponential function declines the relationship. After corrosion of 28 d sandstone specimens, the dynamic compressive strength and dynamic elastic modulus are much lower than the uncorroded specimens. The average strain rate and the dynamic peak strain with the corrosion time extension are a quadratic function of the increasing relationship after the corrosion effect of the sandstone dynamic peak strain, and the average strain rate is significantly greater than the uncorroded specimens. With the corrosion time extension of sandstone specimens by the impact of damage degree being increased, the 14 d and 28 d specimen ruptures' degree is much greater than other time gradients. The dynamic tensile strength of the split specimens decreases with increasing corrosion time; the dynamic peak strain increases quadratically; and the transmitted energy also decreases with increasing corrosion time.

**Keywords:** rock dynamics; alkali corrosion; corrosion time; dynamic mechanical properties; split Hopkinson pressure bar (SHPB)

## 1. Introduction

Currently, China's rapidly developing social economy has an increasing demand for deep resources, energy, and urban underground space [1]. Tunnels, subways, and other underground engineering structures are constantly subjected to various corrosion damages caused by the complex hydrochemical environment around them, which leads to the continuous deterioration of physical and mechanical properties of rock materials, seriously affecting the safety and durability of underground engineering. At the same time, engineering surrounding rock subjected to long-term hydrochemical corrosion damage will

also be subjected to mechanical impact rock breaking, blasting excavation, and earthquake and other dynamic load impacts [2], posing a great threat to the safety and stability of underground engineering.

Shallow rock works are susceptible to acid rain erosion, and many scholars at home and abroad have carried out research on mechanical property tests, intrinsic models, and numerical simulations of rocks under the effect of acidic chemical corrosion. In terms of static tests, Wu et al. [3] used acoustic emission technology to analyze the damage and failure process of limestone under uniaxial compression after being corroded by different acid solutions; Wang et al. [4] conducted uniaxial compression and scanning electron microscopy tests on panel sandstone under the action of different acidic solutions and discussed the corrosion mechanism of panel sandstone under the action of hydrochemicals; Tian et al. [5] carried out triaxial compression tests on marble after immersion in the $Na_2SO_4$ solution under different acidic conditions and compared and analyzed the strength damage, deformation characteristics, and mechanical parameters' response mechanisms of granite under a hydrochemical environment with different pH values; Li et al. [6] conducted indoor compressive strength test and CT scan test of calcareous cemented feldspar sandstone with different pH solutions and proposed a rock chemical damage model that can be applied to acidic solutions to quantitatively describe the process of rock dissolution; Ma et al. [7] conducted dissolution kinetics test and uniaxial compression test on limestone and calcite under acidic hydrochemical solution corrosion and established the dissolution kinetics equation and strength damage equation of limestone and calcite specimens; Chen et al. [8] conducted indoor simulated acid rain solution marble immersion test and uniaxial compression test to study the effect of acid rain corrosion on the uniaxial compression properties of marble. As for the dynamic mechanical tests, Liu et al. [9] conducted static and dynamic load coupling tests on red sandstone immersed in acidic solution for 30 days and analyzed and summarized the fractal dimension and related laws after rock fragmentation; Zhang et al. [10] conducted the dynamic tensile test on limestone specimens corroded by acidic mixed chemical solutions at different times with the separated Hopkins pressure bar and explored the dynamic tensile strength, dissipated energy, and transmission energy of limestone with the change of corrosion damage degree; Liu et al. [11] used the Hopkinson pressure bar test system to conduct a dynamic compression test on marly limestone, silt stone, and pale red limestone specimens after maintenance with different acidic and neutral chemical solutions and investigated the effects of chemical corrosion and chemical corrosion–temperature coupling, chemical corrosion–damage coupling on the dynamic properties of the rocks and their mechanisms.

Many deep rock mass engineering structures both at home and abroad have been subjected to long-term hydrochemical corrosion, such as groundwater [12], which is generally alkaline in nature, and its resulting chemical corrosion, as one of the important factors affecting the long-term stability of underground engineering structures, has a non-negligible weakening effect on the strength, deformation, and damage characteristics of rock masses, which has attracted the attention of scientific researchers at present. In terms of static tests, Hu et al. [13] analyzed the effect of corrosion by different chemical solutions on the strength of sandstone through the uniaxial compression test and came to the conclusion that the stronger the alkalinity of the chemical solutions, the greater the effect on the corrosion of sandstone, namely, the greater the decrease in the strength of sandstone specimens. Wang et al. [14] conducted a triaxial compression test on granite after immersion in different alkaline solutions, and the results showed that the cohesion of the specimens increased with the increase in the pH value of the immersion solution, and the angle of internal friction decreased with the increase in the pH value; Ding et al. [15] conducted a triaxial compression test on limestone after immersion in different pH chemical solutions and studied the characteristics of the effect of chemical corrosion on each stage of the full stress–strain curve of the specimens and found that the pH value was more sensitive to the peak strength of the specimens when it varied from neutral to weak alkali; Xin [16] studied the variation law of macroscopic mechanical properties and macroscopic damage characteristics of weakly

cemented siltstone in different alkaline water environments in the seven mines of Danan Lake and found that the deterioration of various mechanical parameters had an obvious time effect through analysis; Wang et al. [17] conducted different chemical solution immersion tests and triaxial compression tests on malmstone and concluded that the stronger the alkalinity and the higher the concentration of the solution, the stronger the effect on the corrosion of malmstone, and the reduction in the cohesion of the specimens was greater than their internal friction angle. Li et al. [18] analyzed and explored the hydrochemical damage mechanism of sandstone through shear strength tests of sandstone under the action of different hydrochemical solutions. It was found that when the hydrochemical solution was more acidic or alkaline, the secondary porosity was larger, and the change of P-wave velocity was more significant. Cui et al. [19] analyzed and compared sandstone specimens soaked in the NaOH solution for 28 days by using SEM electron microscopy scanning technique and X-ray powder crystal diffraction technique and established a convection–diffusion–reaction model of the water–rock system; the change in porosity was used to quantitatively describe the changes in the micro-structure of the rock due to water–rock action. Wang et al. [20] obtained the creep equation of deep sandstone under chemical corrosion by a uniaxial creep test of sandstone under acidic and alkaline solution corrosion, and the corresponding model fitting curves and creep parameters were obtained by fitting the experimental data and verified its creep model. Yang et al. [21] studied the evolution law of fracture opening in single-fissure granite under the condition of stress-chemical coupling and established the fracture opening evolution model under the permeation of alkaline solution and acidic solution, respectively. As for the dynamic mechanical tests, Li et al. [22] subjected the limestone specimens to corrosion in three groups of solutions with different pH (acid, neutral, and alkaline) and conducted the impact compression tests on them using the separated Hopkinson pressure bar apparatus and found that the porosity of the specimens increased gradually with the increase in corrosion time, and the dynamic compressive strength and dynamic elastic modulus of the limestone showed different degrees of deterioration after corrosion by different solutions, and the fractal dimension also increased.

It can be seen that although there are more studies on the basic physical properties of rocks under the action of chemical corrosion at present, the research results were mainly focused on the static load test of rocks after acidic corrosion, while the dynamic load test of rocks after alkali corrosion was relatively little researched. In practical engineering, some underground projects are subjected to impact loads, such as rock burst and earthquakes, in addition to groundwater corrosion [23]; therefore, it is necessary to study the mechanical properties of rocks after the coupling effect of alkali chemical corrosion and dynamic mechanics.

In order to study the effects of different corrosion times on the physical properties as well as dynamic mechanical properties of sandstone under a strong alkaline solution, sandstone specimens were subjected to the NaOH solution with pH 11 for corrosion (0, 1, 3, 7, 14, and 28 d), and then, the values of their basic physical properties were measured. The impact compression and Brazilian splitting tests were carried out on sandstone specimens after corrosion with alkaline solutions under the same loading conditions using a 50 mm diameter split Hopkinson pressure bar (SHPB) test apparatus. Due to the limitation of space, this paper focuses on the macroscopic mechanical parameters, such as dynamic compressive strength, dynamic stress–strain curve, dynamic elastic modulus, average strain rate, and dynamic peak strain of sandstone with corrosion time in the impact compression test.

## 2. Physical Properties of Sandstone after Different Times of Corrosion by Strong Alkali Solution

The sandstone rock specimens required for the test in this paper were taken from the Dingji coal mine of the Huainan Mining Group, with an off-white appearance (static uniaxial compressive strength was 84.77 Mpa, Young's modulus was 6.41 Gpa). The main mineral composition of the sandstone was quartz, kaolinite, sanidine, and other

components according to the XRD test, and the processed specimens were taken from the same rock to ensure the comparability of the test; the sandstone was processed into a cylindrical specimen with a diameter of 50 mm and a height of 25 mm [24], and the unevenness of the specimen ends was controlled to be less than ±0.02 mm. The non-parallelism of the two ends was less than ±0.05 mm [25], and the axis deviation was less than ±0.25°. A total of 60 specimens were processed, 10 of which were non-corroded, and the remaining 50 were divided into 10 groups for different corrosion time compression and splitting tests, respectively.

In the test, five groups of NaOH solutions with a pH of 11 were placed in glass containers, as shown in Figure 1. Ten specimens were placed into each group of chemical solutions for corrosion at 1, 3, 7, 14, and 28 d. At the end of each time gradient, 10 specimens were taken out of the solution to measure the mass, diameter, length, and other parameters of the specimens again.

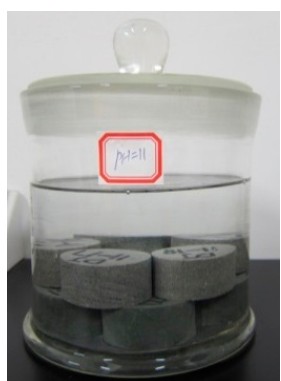

**Figure 1.** Sandstone corroded specimens.

### 2.1. Measurements of Basic Physical Quantities of Sandstone

In this paper, the test set up corrosion at 0 d (non-corroded), 1 d, 3 d, 7 d, 14 d, and 28 d, a total of six time gradients, during the test process, periodically removing the sandstone specimen from the solution, first with a cotton cloth to wipe away the surface moisture, control the surface of the specimen for any dripping liquid, and then letting it stand for 5 min indoors, to ensure that the specimen surface liquid had evaporated dry, and then using the vernier caliper and electronic scales to measure its diameter and length, as well as the mass.

The basic physical parameters, such as volume, mass, and density of sandstone specimens before and after different times of corrosion by the strong alkali solution, are shown in Table 1.

### 2.2. Density and Mass Variation

From Figure 2, it can be seen that the average density of sandstone specimen increases with the corrosion time when the strong alkali solution is corroded for 1 d–14 d, and when the corrosion is 14 d–28 d, the average density growth rate of the sandstone specimen shows a stable trend. Preliminary analysis: 1 d–14 d, the solution enters the rock fissure, resulting in the increase in density of sandstone specimen, while 14 d–28 d, the rock absorbs the solution, gradually tending to saturate; then, the density growth rate tends to be stable at this time.

**Table 1.** Basic physical parameters of sandstone specimens.

| Sample Name | Non-Corroded | | | | Corrosion Time (d) | Corroded | | | |
|---|---|---|---|---|---|---|---|---|---|
| | Mass (g) | Length (mm) | Diameter (mm) | Density (g·cm⁻³) | | Mass (g) | Length (mm) | Diameter (mm) | Density (g·cm⁻³) |
| DJ11-26 | 123.03 | 24.97 | 50.05 | 2.506 | 0 | (non-corroded) | | | |
| DJ11-27 | 126.32 | 25.08 | 50.05 | 2.561 | | | | | |
| DJ11-28 | 127.36 | 25.11 | 50.01 | 2.584 | | | | | |
| DJ11-29 | 127.39 | 25.12 | 50.04 | 2.581 | | | | | |
| DJ11-30 | 126.74 | 25.09 | 50.11 | 2.572 | | | | | |
| DJ11-01 | 127.59 | 25.14 | 50.06 | 2.581 | 1 | 127.87 | 25.12 | 50.12 | 2.582 |
| DJ11-02 | 125.05 | 25.00 | 49.97 | 2.552 | | 125.50 | 24.95 | 50.10 | 2.554 |
| DJ11-03 | 123.36 | 25.10 | 50.04 | 2.500 | | 124.02 | 25.11 | 50.15 | 2.503 |
| DJ11-04 | 127.17 | 25.04 | 50.06 | 2.582 | | 127.47 | 25.03 | 50.12 | 2.583 |
| DJ11-05 | 127.28 | 25.01 | 50.08 | 2.585 | | 127.61 | 25.03 | 50.09 | 2.589 |
| DJ11-06 | 126.89 | 25.03 | 50.01 | 2.582 | 3 | 127.62 | 25.09 | 49.90 | 2.603 |
| DJ11-07 | 123.18 | 24.84 | 50.02 | 2.525 | | 124.09 | 24.94 | 50.14 | 2.530 |
| DJ11-08 | 127.20 | 25.05 | 50.07 | 2.581 | | 127.68 | 25.03 | 50.09 | 2.590 |
| DJ11-09 | 125.12 | 25.10 | 50.03 | 2.537 | | 125.89 | 25.08 | 50.17 | 2.541 |
| DJ11-10 | 127.73 | 24.99 | 50.06 | 2.598 | | 128.23 | 24.98 | 50.15 | 2.601 |
| DJ11-11 | 128.70 | 25.02 | 50.10 | 2.612 | 7 | 129.34 | 25.00 | 50.16 | 2.620 |
| DJ11-12 | 127.81 | 24.97 | 50.00 | 2.608 | | 128.45 | 25.02 | 50.20 | 2.627 |
| DJ11-13 | 124.62 | 25.01 | 50.01 | 2.539 | | 125.68 | 25.12 | 50.15 | 2.544 |
| DJ11-14 | 127.09 | 25.09 | 50.01 | 2.581 | | 127.77 | 25.12 | 50.07 | 2.585 |
| DJ11-15 | 126.81 | 25.10 | 50.09 | 2.565 | | 127.46 | 25.06 | 50.14 | 2.577 |
| DJ11-16 | 128.19 | 25.08 | 50.02 | 2.603 | 14 | 129.05 | 25.09 | 50.11 | 2.610 |
| DJ11-17 | 126.75 | 25.01 | 50.04 | 2.579 | | 127.48 | 24.99 | 50.09 | 2.591 |
| DJ11-18 | 129.14 | 25.08 | 50.04 | 2.619 | | 129.80 | 25.17 | 50.12 | 2.625 |
| DJ11-19 | 125.97 | 25.11 | 50.07 | 2.549 | | 127.07 | 25.11 | 50.16 | 2.562 |
| DJ11-20 | 128.14 | 25.10 | 50.03 | 2.599 | | 128.93 | 25.15 | 50.13 | 2.600 |
| DJ11-21 | 128.48 | 25.12 | 50.05 | 2.602 | 28 | 129.11 | 25.13 | 50.22 | 2.666 |
| DJ11-22 | 126.26 | 25.07 | 49.99 | 2.568 | | 127.12 | 25.11 | 50.10 | 2.570 |
| DJ11-23 | 127.58 | 25.00 | 49.96 | 2.605 | | 128.20 | 25.10 | 50.15 | 2.659 |
| DJ11-24 | 128.14 | 25.03 | 50.03 | 2.606 | | 129.09 | 24.97 | 50.19 | 2.608 |
| DJ11-25 | 127.33 | 25.06 | 50.02 | 2.587 | | 128.07 | 25.12 | 50.07 | 2.591 |

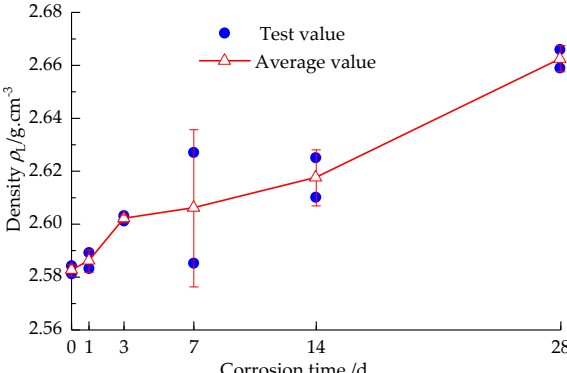

**Figure 2.** Change of the density of sandstone after solution corrosion.

It can be seen from Figure 3 that the sandstone mass changed rapidly after placing the sandstone specimen in strong alkali solution corrosion for 1 d to 14 d. The reason is that when the sandstone specimen is placed in the solution for corrosion, the solution penetrates into the originally existing micro-pores or micro-cracks inside the sandstone, and then, the pores inside the sandstone specimen continue to expand due to chemical reactions, such as hydrolysis, between the aqueous chemical solution and some mineral components in the sandstone, resulting in a secondary micropore. The hydrochemical solution continues to penetrate into these new micropores, resulting in faster growth of the sandstone mass increase rate in the early stage; but, when sandstone was corroded for 14 d to 28 d, the sandstone mass change rate tended to level off. The reason is that the water–rock chemical reaction tended to stabilize, and the absorption solution of the internal micropore fissure in

the sandstone was close to saturation. It can be seen that the hydrochemical corrosion of sandstone has a time effect.

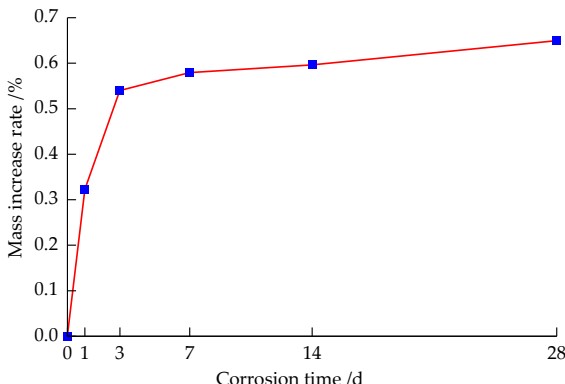

**Figure 3.** Increase rate of sandstone mass after solution corrosion.

The hydrochemical corrosion process of sandstone must be accompanied by the change of some macroscopic physical parameters, among which the significant phenomenon is that the mass of the sandstone specimen changes with the change of corrosion time. Additionally, the mass change of sandstone specimens in the corrosion process is often compared and analyzed with the mass increase rate as a measurement index, which can be calculated as shown in Equation (1).

$$W(t) = (m_t - m)/m \tag{1}$$

where $W(t)$ represents the sandstone specimen after corrosion by strong alkali solution mass increase rate, %; $m_t$ and $m$ are sandstone specimen corrosion times $t$ after and not corrosive mass.

## 3. Dynamic Compressive Mechanical Properties of Sandstone after Corrosion by Strong Alkali Solution

In this paper, the Hopkinson pressure bar (SHPB) apparatus was adopted from the State Key Laboratory of Mining Response and Disaster Prevention and Control in Deep Coal Mines. The SHPB test apparatus consists of five parts: impact loading system, load transfer system, data acquisition system, speed measuring device system, and preload axial compression system [26], as shown in Figure 4.

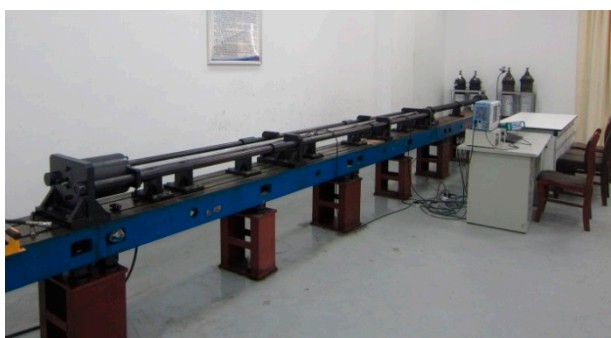

**Figure 4.** Split Hopkinson pressure bar test apparatus.

The rods were machined from high-strength alloy steel with an elastic modulus of 210 GPa and p-wave velocity of 5190 m/s. The incident bar, transmission bar, and absorbing bar were, respectively, 2000, 1500 and 1000 mm in length, with diameters of 50 mm. 120-3AA endless weld-free strain gauges were used for the strain gauges. The impact bar (bullet) adopted the spindle shape to realize the half-sine wave loading and satisfy the stress uniformity of the specimen in the impact process. The impact adopted nitrogen compressed gas. The driving air pressure for the impact compression test and

dynamic splitting test was 0.6 MPa and 0.3 MPa, respectively, and the spindle-shaped bullet was strictly controlled to be in the same position in the launch tube before each impact, so that the impact distance of the impact bar remained unchanged in order to ensure the consistency of the applied impact load, so that the impact velocity and loading waveform obtained during the test with the same driving air pressure remained consistent.

### 3.1. Dynamic Stress–Strain Curves of Sandstone at Different Corrosion Times

The sandstone specimens were subjected to SHPB impact compression tests after corrosion by strong alkali solutions at different times, and the dynamic stress–strain curves were obtained after treatment, as shown in Figure 5.

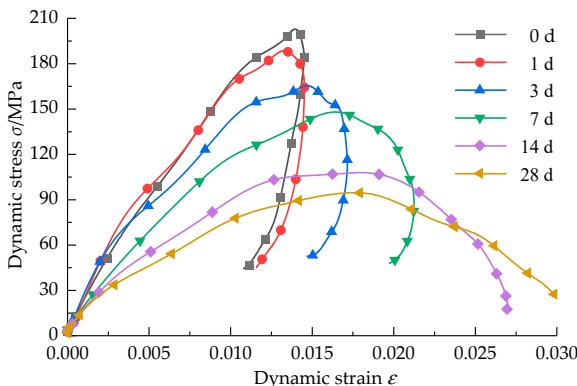

**Figure 5.** Dynamic stress–strain curves of sandstone at different corrosion times.

According to the basic principle of the SHPB test [27], data processing by the three-wave method [28] can be used to obtain mechanical parameters, such as dynamic stresses and strains in sandstone specimens.

The $\sigma$-$\varepsilon$ curve change law is basically the same; the curve length in the elastic deformation stage is larger than that in the plastic deformation stage. When the specimens were non-corroded and corroded at 1 d, the $\sigma$-$\varepsilon$ curve peak front was steeper, and the strain was relatively small from 1 d to 3 d; in the curve of the stage line elastic stage, the slope of the sandstone specimen curve decreased with the increase in corrosion time, that is, the dynamic elastic modulus of the specimen decreased with the increase in corrosion time. In the third stage of the curve, the dynamic peak stress of the sandstone after corrosion at 28 d was much smaller than the other time gradients when loaded, and the corresponding damage strain also increased sharply. Similarly, we can deduce that the dynamic peak stress of the sandstone also had a time effect like the dynamic elastic modulus (with corrosion time extension, the sandstone specimen softening effect will also increase), thus confirming the increased micropore fissures in the sandstone specimen, making the sandstone specimen corrosion softened by corrosion. At the same time, the flexibility became stronger, and the corrosion effect of the alkali solution changed the internal composition and microstructure of the sandstone; with the extension of corrosion time, the length gap between the elastic deformation curve and the plastic deformation decreased. This phenomenon indicates that with the extension of corrosion time, sandstone brittleness gradually weakened, and ductility gradually increased. According to the composition of sandstone in Section 2, we concluded that with the extension of corrosion time, cations such as $K^+$, $Na^+$, $Ca^{2+}$, $Mg^{2+}$ in the sandstone and quartz, the main component of the sandstone, continue to combine with $OH^-$ ions in the solution and undergo chemical reactions, so that the original mineral composition is destroyed by decomposition, secondary fractures increase, and the sandstone structure becomes loose, which is similar to the conclusion of the study by Chen et al. [29] and Wang et al. [14].

### 3.2. Dynamic Compressive Strength and Dynamic Peak Strain Variation

Under the condition of impact air pressure 0.60 MPa, the deterioration of dynamic compressive strength with corrosion time of the sandstone specimens corroded by strong alkali solution at different times is shown in Figure 6.

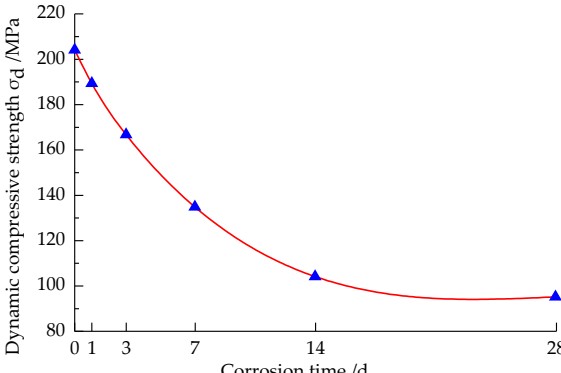

**Figure 6.** Variation curve of dynamic compressive strength with the solution corrosion time.

It can be seen from Figure 6 that the dynamic compressive strength of the standard sandstone specimens without corrosion reached 204.11 MPa, while the dynamic compressive strength of the specimens decreased by 7.27%, 18.36%, 34.06%, 49.14%, and 53.53% after corrosion for five different times, respectively, compared with the non-corroded specimens. It shows that the dynamic compressive strength of the sandstone specimens corroded by strong alkali solution was obviously lower than that of non-corroded specimens, that is, it deteriorated more greatly; the dynamic compressive strength of the sandstone specimens corroded by different times with the extension of corrosion time showed a sharp decline in the early stage and a steady and slow decline in the later stage, and the dynamic compressive strength of the specimens corroded at 28 d was the smallest. We compared our results with those in [1,22] and found that compared with the sandstone specimens in our study, the dynamic compressive strength of the limestone specimens decreased less with the extension of corrosion time after corrosion by a strong alkali solution. The reason for this analysis is that the main component of the limestone is calcite, which is not easily dissolved in a strong alkali solution, and only a small amount of hydrolysis reaction exists, while the main component of sandstone is quartz, which easily reacts with $OH^-$ ions in strong alkali, and with the prolongation of corrosion time, the internal microfracture pores of the sandstone increased, leading to its structural deterioration and causing an obvious decrease in dynamic compressive strength, and the dynamic compressive strength of the specimens decreased in a quadratic function relationship with corrosion time, as shown in Equation (2).

$$\sigma_d = 0.2445t^2 - 10.571t + 199.93 \ (R^2 = 0.9942) \tag{2}$$

where $\sigma_d$ is the dynamic compressive strength of the specimen after corrosion at different times, MPa.

The dynamic peak strain curve of the specimen after corrosion by the strong alkali solution for different times is shown in Figure 7.

As can be seen from Figure 7, the dynamic peak strain of the sandstone specimen increases as a quadratic polynomial function with the extension of corrosion time, and the curve is fitted as shown in Equation (3).

$$\varepsilon_d = -0.006t^2 + 0.3248t + 14.042 \ (R^2 = 0.9983) \tag{3}$$

where $\varepsilon_d$ is the dynamic peak strain of the sandstone after different times of corrosion by the strong alkali solution, $10^{-3}$.

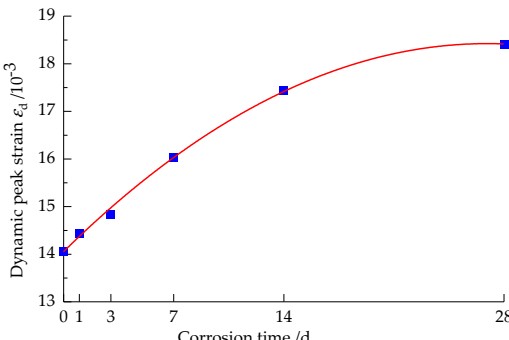

**Figure 7.** Variation curve of dynamic peak strain with the solution corrosion time.

The dynamic peak strain of each sandstone specimen after different times of corrosion by a strong alkali solution is greater than that without corrosion, which shows that the strain-softening effect of sandstone specimens after corrosion placed in a strong alkali solution is significant, and the softening effect of the specimens will be enhanced with the increase in corrosion time. The $\varepsilon_d$ value of sandstone specimens corroded for 1 d increased by 2.60% compared with that of non-corroded specimens. Compared with 1 d, the $\varepsilon_d$ value increased by 2.73% at 3 d. The $\varepsilon_d$ value after 7 d of corrosion increased by 8.13% compared with that after 3 d of corrosion. The $\varepsilon_d$ value after 14 d of corrosion increased by 8.79% compared with that after 7 d, while the $\varepsilon_d$ value after 28 d of corrosion only increased by 5.54% compared with that after 14 d of corrosion, that is, the increase was most significant when the corrosion was 7 d–14 d, and the growth rate of the dynamic peak strain decreased when the corrosion was 14 d–28 d.

From the analysis of the water–rock chemical reaction and sandstone microstructure, the whole reaction process can be divided into two stages. The first stage is the initial stage of reaction, in which the water–rock chemical reaction rate is faster, and the reaction phenomenon is significant; at a micro level, the $OH^-$ ions in the strong alkali solution react with part of the components in the sandstone specimen, and the internal micro porosity expands violently, and the porosity increases sharply, which makes the original dense structure of the rock become relatively loose and fragile gradually. At a macro level, it shows a sharp deterioration of the dynamic compressive strength and a rapid increase in the dynamic peak strain. The second stage is the late stage of reaction, when the reaction between $OH^-$ ions and part of the components of sandstone specimens is close to equilibrium, the chemical reaction rate of water–rock starts to slow down, and the rate of increase in sandstone porosity gradually decreases, and the decrease in the dynamic compressive strength and the increase in the dynamic peak strain also tends to level off.

### 3.3. Variation of the Dynamic Elastic Modulus

The dynamic elastic modulus of sandstone specimens corroded by strong alkali solution at different times is shown in Figure 8.

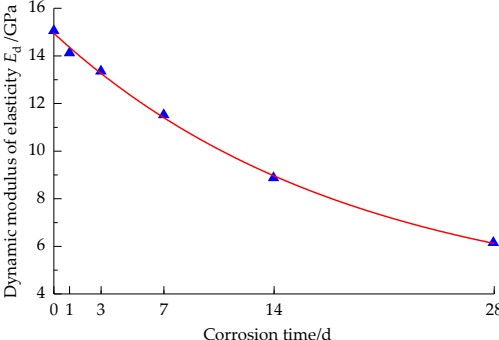

**Figure 8.** Variation curve of dynamic elastic modulus with the solution corrosion time.

It can be seen from Figure 8 that the dynamic elastic modulus of sandstone specimens corroded by the strong alkali solution is smaller than that of non-corroded specimens. The $E_d$ of sandstone specimens shows an overall decreasing trend with the extension of corrosion time, which is consistent with the dynamic compressive strength and reaches the minimum at 28 d of corrosion. The dynamic elastic modulus of the sandstone specimen decreases rapidly with the extension of corrosion time from 1 d to 14 d, while at 14 d–28 d, the rate of decrease in dynamic elastic modulus with the extension of corrosion time becomes smaller and gradually becomes stable, which is similar to the findings of Li et al. [22], but in contrast, under the same corrosion time conditions, the pre-decrease in dynamic elastic modulus of the sandstone specimens under the strong alkali solution corrosion was significantly greater than that of the limestone specimens. By fitting the variation trend of the dynamic elastic modulus of the sandstone, it is found that the dynamic elastic modulus of the sandstone specimen has an exponential term relationship with corrosion time, as shown in Equation (4).

$$E_d = 14.6e^{-0.032t} \quad \left(R^2 = 0.9922\right) \tag{4}$$

where $E_d$ is the sandstone dynamic elastic modulus after different times of corrosion by the strong alkali solution, GPa.

The $E_d$ value loss percentage of corroded specimens was 6.21%, 11.36%, 23.58%, 41.19%, and 59.34%, respectively, compared with that of non-corroded specimens. This indicates that the corrosion treatment of the strong alkali solution results in the obvious deterioration of the dynamic mechanical properties of the sandstone, and it shows a time effect.

### 3.4. Variation of the Average Strain Rate

Figure 9 shows the change curve of the average strain rate of the sandstone specimens after corrosion by the strong alkali solution at different times.

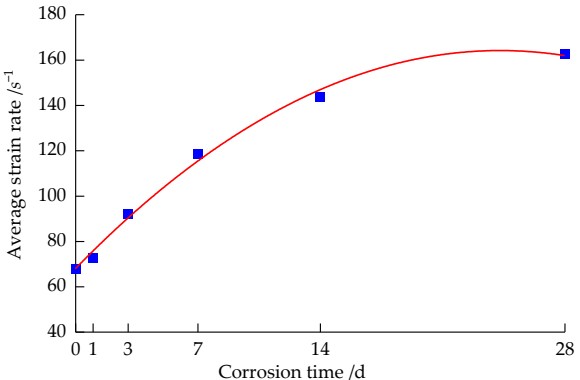

**Figure 9.** Variation of average strain rate with the solution corrosion time.

It can be seen from Figure 9 that the average strain rate of the sandstone specimens increases as a quadratic function of corrosion time, and the curve is fitted as shown in Equation (5).

$$\dot{\varepsilon} = 0.1628t^2 + 7.9236t + 67.783 \left(R^2 = 0.9955\right) \tag{5}$$

where $\dot{\varepsilon}$ is the average strain rate of the sandstone after different times of corrosion by the strong alkali solution, $s^{-1}$.

The average strain rate of the sandstone specimens at 1 d–7 d of corrosion treatment increased rapidly with the extension of corrosion time, while when the specimens were corroded at 7 d–28 d, the growth rate of the average strain rate began to slow down with the extension of corrosion time. On the whole, the average strain rate increased with the extension of corrosion time of the specimens in the strong alkali solution. Typically, the rock strength in the same rock increases with the increasing strain rate [30]. In contrast, the

dynamic compressive strength of the specimens in the tests in this paper decreased with the increasing average strain rate, and the two were negatively correlated, fully demonstrating the time effect of the sandstone specimens under corrosion by the strong alkali solution, i.e., the deterioration of the specimens increased with increasing corrosion time.

### 3.5. Specimen Compression Damage Pattern

As can be seen in Figure 10a, the specimen was in the splitting tensile damage when the stress was small, and the specimens were relatively intact as a whole; compared with Figure 10a, after 1 day of corrosion, the specimens were slightly less intact as a whole, and the number of fragments increased from one to four–five; with the extension of corrosion time, in Figure 10c–e, the fragmentation of the specimens increased, the number of fracture surfaces and fragments increased, and more small-sized fragments were created; comparing Figure 10b–e, it can be observed that with the prolongation of corrosion time, the sandstone specimen as a whole showed a trend of spalling-type damage from the outside to the inside, while in Figure 10f, the specimen was more obviously crushed by the impact crushing effect, and the damage pattern of the specimen shows that there are basically only small-sized fragments and granular debris.

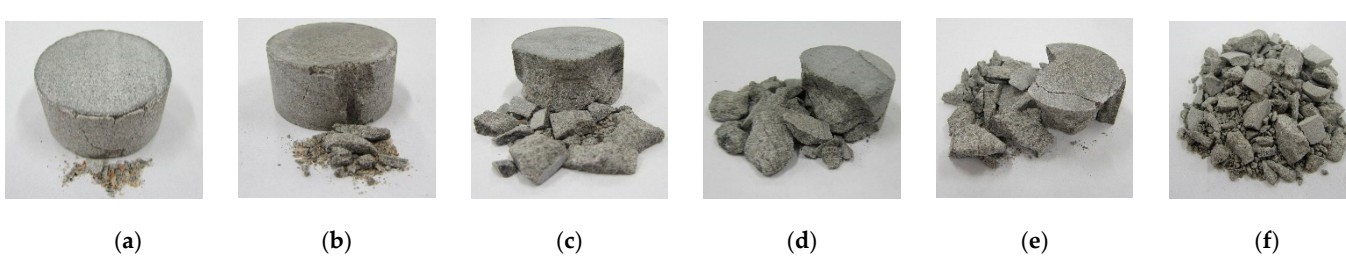

|       |       |       |       |       |       |
|-------|-------|-------|-------|-------|-------|
| (**a**) | (**b**) | (**c**) | (**d**) | (**e**) | (**f**) |

**Figure 10.** Impact compression failure forms of sandstone with different corrosion times. Non-corroded (**a**), corroded 1 d (**b**), corroded 3 d (**c**), corroded 7 d (**d**), corroded 14 d (**e**), and corroded 28 d (**f**).

In terms of the specific consideration of the impact of corrosion, we believe that the reason is that with the extension of corrosion time, quartz, the main component of sandstone, constantly reacts with $OH^-$ ions in the strong alkaline solution [19]:

$$SiO_2 \ (quartz) + 2OH^- \rightarrow H_2SiO_4{}^{2-} \tag{6}$$

In addition, other mineral components present in the sandstone in small amounts constantly react chemically or hydrolytically with $OH^-$ ions to produce new substances, e.g., sanidine reacts readily with water and $OH^-$ ions [31]:

$$KAlSi_3O_8 \ (sanidine) + 2H_2O + 6OH^- \rightarrow Al(OH)_4{}^- + 3H_2SiO_4{}^{2-} + K^+ \tag{7}$$

With the corrosion time extended, the continuous water chemistry dissolves the sandstone gradually, the original dense structure becomes loose, the sandstone's internal secondary pores increase, and the original cracks continue to expand, causing the internal structural deterioration of the sandstone (which is consistent with the results from Refs. [19,22,31]), resulting in the dynamic compressive strength of the specimen also being reduced, so that the degree of damage to the specimen continues to increase, and the longer the corrosion time, the more obvious the increase, which corresponds to the law that their dynamic compressive strength decreases with increasing corrosion time.

### 4. Dynamic Splitting Test of Sandstone after Corrosion by Strong Alkali Solution

The SHPB device was used to conduct dynamic Brazilian splitting tests on sandstone specimens. The Memrecam H-3E high-speed camera produced by NAC Japan was used, and a xenon lamp was used to supplement the light to complete the high-speed video of the

sandstone dynamic splitting test. The high-speed camera and the supporting equipment are shown in Figure 11.

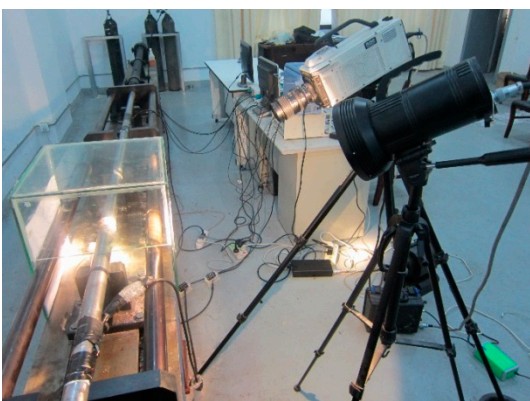

**Figure 11.** High-speed camera and its supporting instruments and equipment.

The dynamic splitting crack development process of the photographed sandstone specimen is shown in Figure 12.

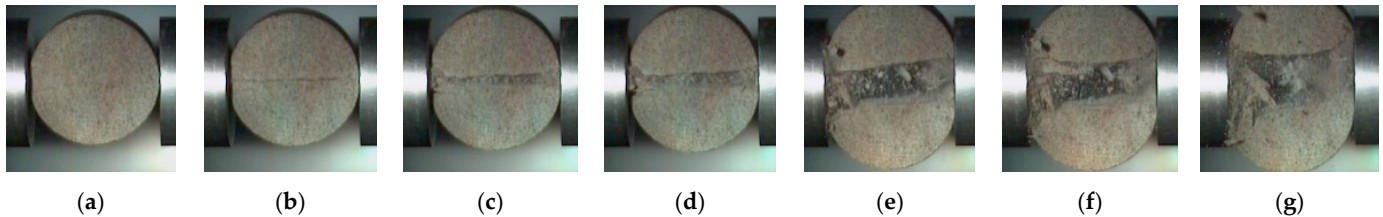

(**a**) (**b**) (**c**) (**d**) (**e**) (**f**) (**g**)

**Figure 12.** Dynamic tensile crack development process in sandstone under high-speed camera. (**a**–**g**) Crack extension pattern with time under high-speed camera.

From Figure 12, it can be seen that the splitting tensile damage morphology of the specimen exhibits radial splitting into two complete parts along the loading, and the radial cracking surface of the sandstone specimen gradually expands with time, with obvious directionality, thus satisfying the validity conditions of the Brazilian disc test [32–34].

## 4.1. Dynamic Tensile Strength and Dynamic Peak Strain Variation

The changes of dynamic tensile strength and dynamic peak strain of sandstone after different times of corrosion by the strong alkali solution are shown in Figure 13a,b, respectively.

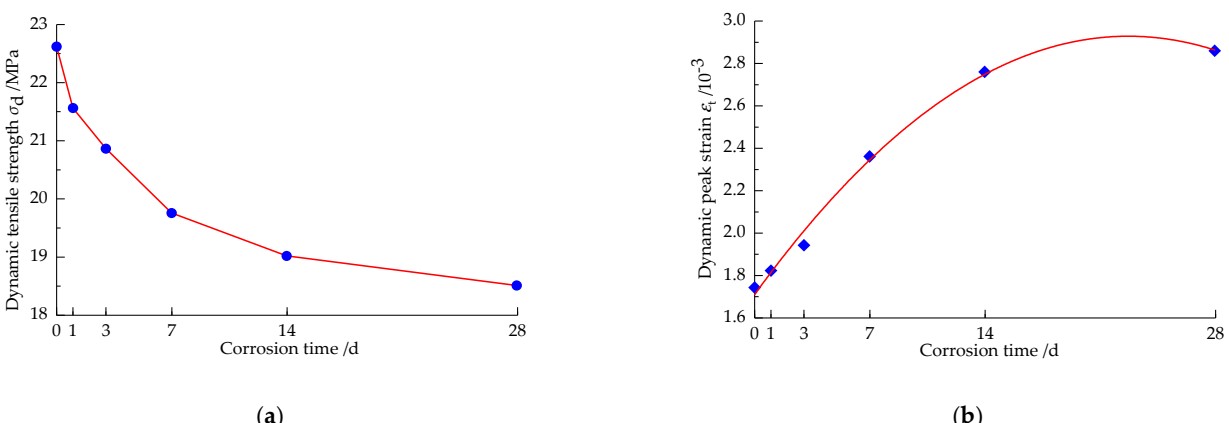

(**a**) (**b**)

**Figure 13.** Changes in dynamic tensile strength and dynamic peak strain of the sandstone after different times of corrosion by the strong alkali solution. (**a**) Dynamic tensile strength. (**b**) Dynamic peak strain.



Figure 13 shows the variation pattern of the dynamic tensile strength $\sigma_t$ and the dynamic peak strain $\varepsilon_t$ of the sandstone specimens with corrosion time in the splitting test. It can be seen that the dynamic tensile strength of the sandstone specimen gradually decreases with increasing corrosion time, while the dynamic peak strain increases quadratically, polynomial with increasing corrosion time. The whole process can be divided into two stages: at 1 d–14 d, as the OH⁻ ions in the strong alkali solution penetrate into the sandstone interior and chemically react with certain mineral components, which leads to the generation of new cracks or the growth of the original cracks, the dense structure of the sandstone gradually becomes relatively loose and fragile, causing a sharp deterioration of the kinetic properties of the specimen, so that the macroscopic dynamic tensile strength of the sandstone specimen decreases more rapidly, from 22.62 MPa to 19.01 MPa, decreasing by 15.95%, while the dynamic peak strain increases more rapidly, increasing by 59.07%. At 14 d–28 d, the strong alkali solution continues to produce a small amount of chemical reaction with some components in the sandstone, and the dynamic tensile strength decreases to 18.50 MPa compared with the previous reaction stage, with a smaller decrease; the dynamic peak strain increases to 2.86, with the same small increase. Throughout the corrosion process, the dynamic tensile strength of the sandstone specimens decreased by 18.23%, and the dynamic peak strain increased by 64.42%. Overall, the increase in the internal damage of the sandstone caused by chemical action resulted in a significant deterioration of its dynamic tensile strength. The fitted relationship for the dynamic peak strain $\varepsilon_t$ of the split sandstone specimens after different times of strong alkali corrosion is shown in Equation (8).

$$\varepsilon_t = -0.0024t^2 + 0.1085t + 1.7043 \ (R^2 = 0.9950) \tag{8}$$

### 4.2. Specimen Splitting Damage Pattern

The impact of rupture pattern of sandstone specimens after different corrosion times is shown in Figure 14.

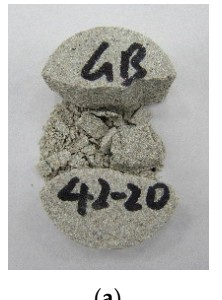 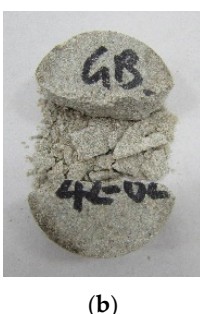 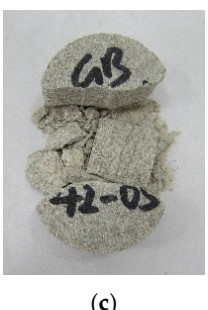 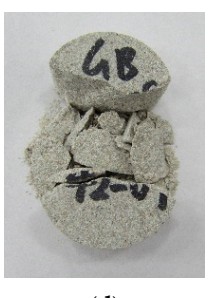 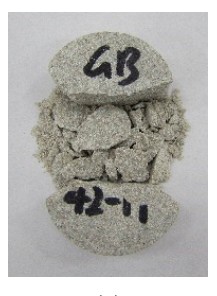 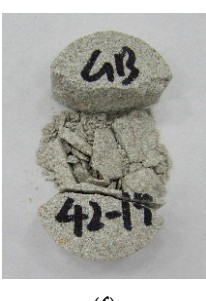

| (a) | (b) | (c) | (d) | (e) | (f) |

**Figure 14.** Dynamic tensile damage forms of sandstone with different corrosion times. (**a**) 0 d (**b**) 1 d (**c**) 3 d (**d**) 7 d (**e**) 14 d (**f**) 28 d.

From Figure 14a–c, it can be seen that the number of specimen fragments is small when the specimen from 1 d to 3 d of immersion, the sandstone specimen splitting damage of the more complete two parts of the area change, is not very obvious. While comparing Figure 14d–f, two end faces at the local crushing area and the local crushing area with the corrosion time extension gradually increase. In addition, in the immersion at 7 d–28 d, the specimen splitting damage form is shown. The specimen damage degree increases, and the number of small-sized fragmentations with the extension of corrosion time increase.

Combined with the above analysis, we can preliminarily conclude that with the prolongation of corrosion time, the specimen internal microporosity and fissures increase, producing damage at the microstructural level, before the main crack penetration, micro cracks under the action of the impact of dynamic load expansion, and then, the formation of multiple rupture surfaces. The damage area increases, and the number of secondary cracks increases. With the aggravation of corrosion, the load-bearing capacity of the specimen also decreases sharply, and the size of the fragment is subsequently reduced, and the area of

the crushed area increases, which is mainly manifested in the macroscopic reduction in the dynamic tensile strength, and the conclusions obtained in Section 4.1 should be confirmed.

### 4.3. Specimen Energy Analysis

In order to better understand the influence of corrosion time on the process of sandstone deformation damage from the perspective of energy evolution, the variation law of transmitted energy of sandstone in dynamic impact deformation damage with corrosion time is studied. Since the incident shock air pressure is kept constant, the incident energy and dissipation energy are not further explored, where the incident wave energy $E_I$ reflects the wave energy $E_R$ and transmits the wave energy $E_T$ of the rock specimen, as shown in the relationship in Equation (9).

$$\left. \begin{array}{l} E_I = A_0 C_0 E_0 \int_0^\tau \varepsilon_I^2(t)\mathrm{d}t \\ E_R = A_0 C_0 E_0 \int_0^\tau \varepsilon_R^2(t)\mathrm{d}t \\ E_T = A_0 C_0 E_0 \int_0^\tau \varepsilon_T^2(t)\mathrm{d}t \end{array} \right\} \tag{9}$$

where $A_0$ is the cross-sectional area of the compression bar; $E_0$, $C_0$ are the moduli of elasticity of the compression bar material and the longitudinal wave velocity; $\varepsilon_I(t)$, $\varepsilon_R(t)$, $\varepsilon_T(t)$ are the incident, reflected and transmitted stress waves at time $t$, respectively, and the compressive stress is taken as the positive direction; $t$ is the duration of the stress wave.

The variation curve of the ratio of transmitted energy to incident energy with corrosion time is shown in Figure 15.

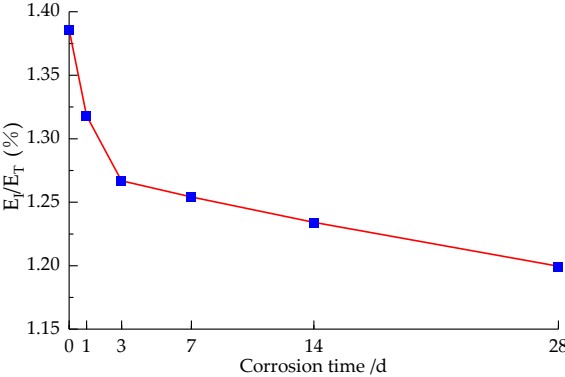

**Figure 15.** Transmitted energy percentage change curve with the corrosion time.

It can be seen that, in the corrosion at 1 d–14 d, the ratio of $E_T$ to incident energy of the specimen decreased from 1.39% to 1.23%, the ratio of transmitted energy corresponding to the uncorroded specimen decreased by 10.98%, and the change in this stage is more obvious; meanwhile, in the corrosion at 14 d–28 d, the ratio of transmitted energy decreased from 1.23% to 1.20%, corresponding to the uncorroded specimen decreasing by 13.48%, the change being relatively flat. The preliminary analysis of the reason is that with the prolongation of corrosion time, the specimen within the micro-cracks continues to sprout and expand, and the macroscopic performance of resistance to deformation deterioration and the transmitted wave intuitively reflects the dynamic stress changes in the specimen. In Section 4.1, it can be deduced from the specimen that the dynamic tensile strength with the prolongation of corrosion time is significantly reduced, so with the prolongation of corrosion time, the transmitted energy ratio decreases, and at 14 d–28 d, the transmitted energy ratio decreases smoothly, and the dynamic tensile strength decreases correspondingly with the law. At the same time, this is corroborated in Section 4.2, where the extent of sandstone damage increases with corrosion time.

## 5. Conclusions

At present, no substantial research has been conducted at home or abroad on the effect of corrosion of strong alkali solutions on sandstone dynamics, which is a new direction of this paper to test the research on the mechanical stability of rocks in engineering.

In this paper, sandstone specimens were subjected to corrosion treatment for different time periods using a strong alkali solution with a pH value of 11. Afterward, the SHPB impact compression tests and Brazilian splitting tests were conducted on uncorroded and corroded sandstone specimens under the same air pressure conditions, and the physical and dynamic mechanical properties of sandstone specimens subjected to the strong alkali corrosion process were analyzed, and the following conclusions were drawn.

(1) In the early stage of corrosion, the reaction between the solution and some components of the sandstone led to the continued expansion of the pores inside the sandstone specimen, as well as the generation of new fissures into which the solution penetrated, resulting in a faster increase in the mass increase rate and average density of the sandstone and a stabilization of the water–rock chemical reaction in the later stage, resulting in a slower increase in the mass increase rate and average density of the sandstone.

(2) In the impact compression test, after different corrosion times, the sandstone dynamic stress–strain curve change law was basically the same, but the shape was different. As the corrosion time increased, the length gap between the linear elastic deformation phase curve and plastic deformation phase curve became smaller. The phenomenon indicated that with the corrosion time increasing, sandstone brittleness gradually weakened, and ductility gradually increased.

(3) With the corrosion time extended, the sandstone specimens' dynamic compressive strength and dynamic elastic modulus significantly deteriorated, both of them showing two stages of rapid decline from 1 d to 14 d and a lowdown from 14 d to 28 d. The dynamic peak strain of the sandstone specimens increased as a quadratic function with the extension of corrosion time. With the corrosion time at 1 d–14 d, the sandstone specimens' $\varepsilon_d$ value increased faster, and at 14 d–28 d, the $\varepsilon_d$ value growth rate had a decreasing trend. The sandstone average strain rate $\dot{\varepsilon}$ and corrosion time were a quadratic positive correlation. With corrosion time increasing, the specimen impact damage degree increased $\dot{\varepsilon}$.

(4) In the Brazilian splitting test, the dynamic properties of the specimens deteriorated continuously due to the corrosion of the strong alkali solution with the prolongation of corrosion time. The transmitted energy and dynamic tensile strength of sandstone specimens both showed a trend of faster decline in the early stage and a slower decline in the later stage with the prolongation of corrosion time; the dynamic peak strain of the sandstone specimens showed a quadratic polynomial positive correlation with the prolongation of corrosion time.

**Author Contributions:** Conceptualization, K.S.; Data curation, C.W.; Funding acquisition, Q.P.; Methodology, Q.P.; Software, C.W.; Supervision, Q.G., K.S., Y.W. and S.W.; Validation, S.S.; Visualization, Q.G. and Y.W.; Writing—Original Draft, Q.P. and C.W. All authors have read and agreed to the published version of the manuscript.

**Funding:** This research was funded by the National Natural Science Foundation of China (No. 52074005, No. 52074006), Anhui Provincial Natural Science Foundation (No. 1808085ME134), and Anhui Postdoctoral Science Foundation (No. 2015B058), Anhui University of Science and Technology Graduate Innovation Fund Project (No. 2021CX2032).

**Institutional Review Board Statement:** Not applicable.

**Informed Consent Statement:** Informed consent was obtained from all subjects involved in the study.

**Data Availability Statement:** The data used to support the findings of this study are available from the corresponding author upon request.

**Acknowledgments:** Thanks are due to the State Key Laboratory of Mining Response and Disaster Prevention and Control in Deep Coal Mine, Engineering Research Center of Underground Mine Construction, Ministry of Education, and Anhui University of Science and Technology, for providing the experimental conditions.

**Conflicts of Interest:** The authors declare no conflict of interest. The funders had no role in the design of the study; in the collection, analyses, or interpretation of data; in the writing of the manuscript, or in the decision to publish the results.

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
