# Peer review of "Experimental Study on Dynamic Mechanical Properties of Sandstone Corroded by Strong Alkali"

_applsci, doi:10.3390/app12157635_

Round 1

Reviewer 1 Report

Article presents results of the reasearch on sandstone samples subjected to corrosion in NaOH. Authors performed tests, using the split Hopkinson pressure bar (SHPB) technique and obtained, among others, stress-strain curves for different corrosion times. Additionaly change in the samples density and mass was controled during the test.

Article is written in the logical way, the language is easy to understand. However, there are several problems:

1) Main problem is fact, that authors presented results from only one test - SPHB. In the case of studying the change of the mechanical properties of the sandstone additional tests should be performed, such as uncofined compression test, Brazillian test and tri-axial compression test. Only after carrying out such a wide range of tests we can fully determine the influence of a factor (in this case, the effect of NaOH) on the mechanical properties. In its current form, the manuscript presents only results of a standard, very often performed test, which is performed on samples made of sandstone, a rather well-known material. Thus the novelty factor of the manuscript is very low.

2) In table 1, the density of the corroded samples is presented. What is the standard deviation of the measurements?

3) In my opinion the standard deviation of the samples density should be included on figure 3. 

4) In what way were the average values of the samples density (presented in fig.3) determined? Because from the table 1 we can calculate, that average density of the 14-day corroded sample is 2,6188 g/cm-3 and of the 28-day corroded is 2,5976 g/cm-3. In fig. 3  the values for 14-days and 28-days are swapped.

Reviewer 2 Report

  1. Section 3.1: For basic properties of samples, the uniaxial compressive strength, Young`s Modulus, Poisson`s ratio, etc. under static loading condition should be provided.
  2. Sections 3.3, 3.4, 3.5, 3.7: The authors found the effect of chemical weathering on the mechanical properties of rock, however, the most of statements for the results were made by the author. Is there any comparison between previous studies? The results from this study should be compared and discussed with the previous studies. Please add more discussions with references, for example, opposite results, agreement with the previous study, the detail reason for the specific result.   
  3. (3), (4), (5), (6): The reviewer thought that the fitting model is should be expressed by log or exponential term rather than second-order polynomial equations.
  4. Lines 350-355: The statements should be revised. The general trend in increase of strength with strain rate is only for the same rock specimen, and the trend is different for the rock types. Also, it is affected by the range of strain rate. Any preliminary tests were done to check the strain rate dependency (Dynamic increasing factor) on strength of this sandstone?
  5. The main concern of the reviewer is that the test condition (strain rate) seems to be not consistent for all cases. In the regard, the test results properly indicated the effect of corrosion on the mechanical behavior of rock?  
  6. Figure 10: caption needs to be revised. Figure 2 -> Figure 10. It is difficult to find the differences among the cases in terms of failure pattern, in particular, what is the difference between (a) and (b)? (c) and (d)? (e) and (f)? Furthermore, For Figure 10(a), which failure pattern can be observed from the figure? The review recommend that the figures would be changed to clearly show the effect of corrosion time on the failure pattern, and add some schematic drawing to explain the differences.     
  7. Section 3.7: the results and discussion were not clear in the current manuscript. Just the decrease of compressive strength affected to the failure pattern of rock? Please discuss on the results in detail with the view of the effect of corrosion.
  8. Line 361, “the failure mode of sandstone specimen is vertical tensile failure”: any references or additional observation to support the statement?
  9. Any results for the poisson`s ratio? If there is, please add the results.

Round 2

Reviewer 1 Report

I would like to thank the authors for the response to my comments. 

Comment 1: From the evaluated standard deviation and from Fig.1 we can conclude that we can hardly speak about the change of the sandstone  density. We can draw a straight line going through every corrosion time point and still be within the standard deviation, thus I can not agree with the point (1) from "Conclusions" sections. In my opinion a much more accurate method of density measuring should be used, than an vernier caliper and electronic scales. Additionally, what was the accuraccy of the measurening equipment along with the uncertainty of measurement?

Comment 2: Author stated that a additional tests were carried out but due to the manuscript space limitation they were not included in the text. In my opinion some part of the current text could be shortened (i.e. introduction) in order to present result of those tests. Additionally, as far as I know, the "Applied Sciences" journal has not strict manuscript size limitation and text well over 25 pages has been published.

Comment 3: Conlusions presented by the authors are rather obvious and even authors stated that in point (3) and (4), thus the novelty of the article is quite low. 

Summing up, the authors adressed my comments and changed few technical problems with the manuscript. However, the novelty of the text has not changed and in my opinion is too low to recommend an article for publication in its current form. In my opinion additional test must be included in the manuscript in order to publish it.

Reviewer 2 Report

All comments and suggestions were appropriately addressed, and the manuscript has been improved. No more comments on the submission. However, it is better to proofread the manuscript to correct grammatical errors by a professional editor. I think that the authors can deal with the correction work during the final editing. 

Author Response

Thanks for the reviewer's careful review and high evaluation of this paper. We checked and revised the article again, and touched up the sentences. Looking forward to getting good news about this article.

For details of the changes, see the Abstract in the revised manuscript; the fourth and fifth paragraphs of Section 1; the first paragraph of Section 2.1; the first paragraph of Section 3.2; the second paragraph of Section 3.4; the first paragraph of Section 3.5; the third paragraph of Section 3.6; Section 3.7; and the Conclusions.

All changes in the revised manuscript are marked in blue on a yellow background.